# Aurora A and AKT Kinase Signaling Associated with Primary Cilia

**DOI:** 10.3390/cells10123602

**Published:** 2021-12-20

**Authors:** Yuhei Nishimura, Daishi Yamakawa, Takashi Shiromizu, Masaki Inagaki

**Affiliations:** 1Department of Integrative Pharmacology, Graduate School of Medicine, Mie University, Tsu 514-8507, Japan; tshiromizu@med.mie-u.ac.jp; 2Glocal Research Center for Advanced Medical Science, Mie University, Tsu 514-8507, Japan; minagaki@doc.medic.mie-u.ac.jp; 3Department of Physiology, Graduate School of Medicine, Mie University, Tsu 514-8507, Japan; dyama@doc.medic.mie-u.ac.jp

**Keywords:** primary cilium, aurora kinase A, AKT kinase, trichoplein, lipid raft, proliferation, differentiation, cancer, obesity, ciliopathy

## Abstract

Dysregulation of kinase signaling is associated with various pathological conditions, including cancer, inflammation, and autoimmunity; consequently, the kinases involved have become major therapeutic targets. While kinase signaling pathways play crucial roles in multiple cellular processes, the precise manner in which their dysregulation contributes to disease is dependent on the context; for example, the cell/tissue type or subcellular localization of the kinase or substrate. Thus, context-selective targeting of dysregulated kinases may serve to increase the therapeutic specificity while reducing off-target adverse effects. Primary cilia are antenna-like structures that extend from the plasma membrane and function by detecting extracellular cues and transducing signals into the cell. Cilia formation and signaling are dynamically regulated through context-dependent mechanisms; as such, dysregulation of primary cilia contributes to disease in a variety of ways. Here, we review the involvement of primary cilia-associated signaling through aurora A and AKT kinases with respect to cancer, obesity, and other ciliopathies.

## 1. Introduction

The human genome encodes 538 protein kinases [1] that play crucial roles in cellular homeostasis through both catalytic and noncatalytic mechanisms [2,3]. Kinase activity is subject to environmental and spatiotemporal regulation, and the consequences of protein phosphorylation are thus affected not only by the particular protein substrate but also by the cellular and subcellular context [4,5,6]. Therapeutic targeting of kinases most often focuses on their catalytic activity, but their noncatalytic functions are also essential to cellular homeostasis and are of increasing therapeutic interest. This is particularly true for the main focuses of this review: the serine/threonine kinases aurora kinase A (AURKA) and v-akt murine thymoma viral oncogene homolog (AKT) [4,5,6,7,8].

AURKA, which is itself regulated by phosphorylation and dephosphorylation, plays an essential role in mitosis through its noncatalytic function as a binding partner for several key regulatory proteins. Not surprisingly, dysregulation of this AURKA noncatalytic function is known to contribute to the development and progression of various diseases, including cancer and obesity. One example is the interaction between AURKA and microtubule nucleation factor targeting protein for xenopus kinesin-like protein 2 (TPX2) during mitosis [9]. In melanoma cells, a mutation in a key AURKA-inactivating phosphatase results in aberrant AURKA activation and its sustained interaction with TPX2 [10]. Inhibition of the AURKA–TPX2 interaction has thus been proposed as a novel therapeutic approach for melanoma with minimal effects on the proliferation of normal cells [11,12,13,14]. AURKA also acts as a scaffold protein and stabilizes the oncogenic transcription factor v-myc avian myelocytomatosis viral oncogene neuroblastoma-derived homolog (MYCN) by interfering with its degradation [15]. MYCN is amplified in some forms of neuroblastoma and prostate cancers and predicts poor prognosis [16,17]; therefore, inhibition of AURKA–MYCN binding might also be a useful therapeutic approach in these cancers [13,18,19].

Another clinically important kinase with crucial catalytic and noncatalytic functions is AKT. AKT lies at the crossroads of many interconnecting pathways with functions in cell metabolism, proliferation, differentiation, and survival. In addition, the conformation, but not the catalytic activity, of the AKT kinase domain plays a role in controlling the ability of the adjacent pleckstrin homology (PH) domain to bind membrane-associated lipids [20]. Thus, both catalytic and noncatalytic activities play a role in AKT function [6,21] and, similar to AURKA, AKT dysregulation plays a role in many major disorders, including cardiovascular and metabolic diseases and cancer. Taken together, these observations suggest that therapeutic approaches that specifically target kinase catalytic and noncatalytic functions in a context-dependent manner could not only improve efficacy but also minimize potential adverse effects [21,22,23,24,25,26,27].

Primary cilia are nonmotile, 1–10 μm long antenna-like structures that extend externally from the plasma membrane of a variety of vertebrate cells [28,29,30,31,32,33,34]. The cilium contains a scaffold of microtubules, the axoneme, that transports molecules into and out of the ciliary body through a process known as intraflagellar transport. The axoneme is anchored to the plasma membrane via a basal body that is derived from the mother centriole containing nine circularly arranged triplets of microtubules [29]. Primary cilia contain receptors and channels that detect signals from the extracellular milieu, such as chemical stimulation and mechanical flow, and transduce them into the cell to regulate physiological functions [28,29,30,31,32,33,34,35,36,37,38].

Although primary cilia are nonmotile, cilia formation is dynamically regulated in response to various stimuli [29,30,31,32,33,34,37,38,39,40,41,42]. For example, primary cilia in serum-deprived fibroblasts and retinal pigment epithelial (RPE) cells undergo disassembly upon addition of serum [43,44,45,46]. Furthermore, forced ciliation has been shown to disrupt progression of the cell cycle in proliferating human RPE cells [47,48,49,50], illustrating the importance of these organelles for proper cell function. The abundance of primary cilia is reduced in a range of cancer types, including glioblastoma [51], esophageal squamous cell carcinoma [52], colon cancer [53], cholangiocarcinoma [54,55], pancreatic ductal adenocarcinoma [56,57], clear cell renal cell carcinoma [58,59,60,61], epithelial ovarian cancer [62,63], luminally derived breast cancer [64], prostate cancer [65,66], melanoma [67,68], and chondrosarcoma [69,70], highlighting the importance of negative regulation of the cell cycle and proliferation by primary cilia [30,31,32,33,71,72,73,74,75,76].

In addition to cancer, dysregulation of primary cilia is associated with obesity [77,78,79]. The arcuate nucleus of the hypothalamus is composed of different types of ciliated neurons, including anorexigenic and orexigenic neurons, which express leptin receptors in the primary cilia [80]. Upon binding of leptin to the receptors, production of anorexigenic and orexigenic neuropeptides is increased and decreased, respectively, leading to appetite suppression [81,82]. Accordingly, loss of primary cilia in these neurons impairs the negative-feedback system crucial to controlling appetite [83,84]. Anorexigenic and orexigenic neuronal axons project into second-order neurons in the paraventricular nucleus that express melanocortin 4 receptor (MC4R), a common receptor for anorexigenic and orexigenic neuropeptides, and adenylate cyclase 3 (ADCY3) in the primary cilia [85]. Mutations in MC4R and ADCY3 have been associated with an increased risk of obesity and Type 2 diabetes [86,87,88]. Increased adipogenesis is another important cause of obesity [89]. For example, knockdown of the causative genes for Bardet–Biedl syndrome (BBS), a ciliopathy inherited in an autosomal-recessive manner, suppresses the formation of primary cilia in preadipocytes and increases adipogenesis through activation of peroxisome proliferator activated receptor γ, a master regulator of adipogenesis [90,91].

These examples serve to illustrate how breakdown in the normal physiological regulation of cilia dynamics and signal transduction can contribute to human diseases [28,33,34,92] and further highlight the role of numerous kinases in regulating primary cilia function [63,93,94]. In the remainder of this review, we focus on the association between AURKA (Section 2) and AKT (Section 3) and primary cilia function and how their dysregulation contributes to ciliopathies such as cancer and obesity (Figure 1).

## 2. Aurora Kinase a Signaling and Its Regulation in Primary Cilia

AURKA is a member of the aurora kinase family that play essential roles in regulation of the cell cycle [7,95,96,97,98]. During mitosis, AURKA is activated by autophosphorylation in a manner dependent on its interaction with distinct proteins at different stages of the mitotic process: polo-like kinase 1 (PLK1) and protein aurora borealis at G2/M [99,100], Ajuba LIM protein at prophase [101], and TPX2 at metaphase [102]. Activated AURKA stimulates mitotic entry and centrosome separation and maturation at G2/M [99,100], formation of the microtubule-organizing center, mitotic spindle organization, and chromosome alignment at M phase [103,104]. AURKA also plays important roles during G1 through promoting the disassembly of primary cilia [22,23,30,31,32,33,38,46,47,48,49,50,71,105,106]. Notably, formation of primary cilia is suppressed in several cancers in which expression of AURKA is increased, including epithelial ovarian cancer [63], prostate cancer [66], pancreatic ductal adenocarcinoma [57,107], and glioblastoma [108,109]. These findings suggest that inhibition of AURKA may suppress the proliferation of these cancer cells by promoting ciliogenesis [22,30,31,32,33,34,38,71,74]. Various proteins have been identified as modulators of AURKA-mediated disassembly of primary cilia during G1, including trichoplein (TCHP), neural precursor cell expressed developmentally downregulated 9 (NEDD9), and centrosomal protein 55 (CEP55), which we discuss in more detail here (Figure 2).

### 2.1. TCHP

TCHP, originally identified as a keratin-binding protein [110,111], is a centriolar protein that suppresses the formation of primary cilia by directly interacting with AURKA [27,47,48,49,50,79]. The N-terminal 130 residues of TCHP are essential for its centriolar localization, its interaction with AURKA, and its involvement in the suppression of ciliogenesis [47]. Knockdown (KD) of TCHP in human RPE cells cultured in the presence of serum inhibits AURKA activation, induces ciliogenesis, and suppresses cell proliferation [47]; however, the effects on ciliogenesis and proliferation are suppressed by co-KD of intraflagellar transport 20 (IFT20), a protein required for primary cilia assembly [47,112]. These findings clearly demonstrate that ciliogenesis can inhibit the cell cycle and that TCHP–AURKA regulate cell proliferation via their effects on primary cilia [30,31,32,33,34,38,74] (Figure 2A).

TCHP expression is regulated by the ubiquitin–proteasome system [48,49,50]. TCHP is ubiquitinated by an E3 ligase complex composed of cullin 3, ring-box 1, and potassium channel tetramerization domain-containing 17 (CRL3^KCTD17^) [48]. Conversely, deubiquitination of TCHP is mediated by ubiquitin-specific peptidase 8 (USP8) [50]. KD of KCTD17 and USP8 suppresses and promotes, respectively, the formation of primary cilia in human RPE cells [48,50]. In zebrafish, knockout (KO) of kctd17 impairs ciliogenesis in Kupffer’s vesicle and induces situs inversus [74], whereas KO of usp8 increases ciliogenesis in the pronephric duct and causes renal cysts [50]. Dysregulation of AURKA is also associated with situs inversus and polycystic kidney in mice [113,114]. Thus, impairment of the TCHP–AURKA interaction may contribute to the pathophysiology of these disorders.

The activity of CRL3^KCTD17^ in human RPE cells is unaffected by the presence or absence of serum in the culture medium [48], whereas USP8 activity is stimulated by several serum factors, including epidermal growth factor (EGF), platelet-derived growth factor, and fibroblast growth factor, each of which triggers phosphorylation of USP8 at tyrosine (Tyr) 717 and Tyr810 [50]. Activated USP8 stabilizes TCHP by suppressing its proteasomal degradation, which results in activation of AURKA, suppression of ciliogenesis, and stimulation of RPE cell proliferation [50]. KD of the EGF receptor (EGFR) in human RPE cells inhibits USP8 Tyr717 and Tyr810 phosphorylation, which enables TCHP and AURKA degradation and reverses the serum-induced effects on ciliogenesis and cell proliferation [50]. Notably, simultaneous KD of the EGFR and either IFT20 or centrosomal protein 164, both of which are indispensable for ciliogenesis, antagonizes the effects of EGFR KD on ciliogenesis and proliferation [50], supporting the hypothesis that primary cilia act as brakes on cell proliferation [30,31,32,33,34,38,74].

NDE1-like 1 (NDEL1), a modulator of dynein activity localized at the subdistal appendage of the mother centriole [115,116], is also involved in TCHP regulation [49]. In human RPE cells, NDEL1 is stabilized in the presence of serum and suppresses CRL3^KCTD17^-mediated ubiquitination of TCHP, resulting in activation of AURKA [49]. The mechanism of NDEL1 stabilization is unknown but is likely to involve inhibition of degradation, since NDEL1 undergoes proteasomal degradation in the absence of serum [49]. Interestingly, Ndel1-hypomorphic mice display increased ciliation in kidney tubular epithelial cells [49]. Given that usp8 KO in zebrafish also increases ciliation in the pronephric duct [50], these findings suggest that increased degradation of TCHP and subsequent inhibition of AURKA may be involved in the ciliation and cystic kidney defects observed in animals with NDEL1 or USPS8 KD.

### 2.2. NEDD9

NEDD9, also known as human enhancer of filamentation 1, is another scaffold protein that affects both AURKA activation and primary cilia formation [46]. Binding of NEDD9 to AURKA, which appears to involve NEDD9 serine (Ser) 296 and the N-terminal domain of AURKA [117,118], suppresses proteasomal degradation of AURKA promoted by the ubiquitin ligase anaphase-promoting complex/C (APC/C) in MDA-MB-231 cells, a human breast cancer cell line [118].

NEDD9–AURKA signaling is regulated by several mechanisms (Figure 2B). In primary cilia in human RPE cells, binding of Frizzled receptor to its ligand Wnt family member 5A (WNT5a) stimulates a number of downstream signaling events, including activation of casein kinase 1ε (CK1ε), which phosphorylates disheveled 2 segment polarity protein 2 (DVL2) and induces formation of a complex with PLK1 and SMAD family member 3 (SMAD3) [106]. The DVL2–PLK1–SMAD3 complex inhibits APC10-induced proteasomal degradation of NEDD9, leading to activation of AURKA signaling and promotion of primary cilia disassembly [106]. Calmodulin also stimulates ciliary disassembly by increasing the interaction between NEDD9 and AURKA [105,119]. In the human esophageal squamous cell carcinoma line EC9760, peroxiredoxin 1, an antioxidant protein frequently overexpressed in tumors [120,121], increases NEDD9 expression and AURKA phosphorylation, resulting in suppression of primary cilia assembly [52]. Hyperactivation of NEDD9–AURKA signaling may thus be involved in oncogenesis through suppression of primary cilia formation [122].

Dysregulation of NEDD9–AURKA signaling is involved in several ciliopathy phenotypes. For example, cystogenesis is more extensive in mice with KO of both Nedd9 and polycystin 1 transient receptor potential channel interacting (Pkd1), a causative gene for autosomal dominant polycystic kidney disease, than in mice with KO of Pkd1 alone [123]. The mechanism of elevated cystogenesis in these mice is thought to involve a failure of AURKA activation [123]. Tetratricopeptide repeat domain 8 (TTC8) is involved in formation of primary cilia, and mutation of the *TTC8* gene has been associated with nonsyndromic retinitis pigmentosa [124,125,126]. TTC8 is a member of a protein family associated with BBS [127]. A complex composed of TTC8, BBS6, and inversin stimulates ciliogenesis via suppression of NEDD9–AURKA signaling in human RPE cells [128]. Thus, mutation of BBS8 may contribute to the vision impairment associated with retinitis pigmentosa by alleviating the inhibition of NEDD9–AURKA signaling, resulting in suppression of ciliogenesis.

### 2.3. CEP55

Mutation of the *CEP55* gene is associated with multinucleated neurons, anhydramnios, renal dysplasia, cerebellar hypoplasia, and hydranencephaly (MARCH), a lethal autosomal-recessive fetal ciliopathy [129,130,131]. CEP55 stabilizes AURKA by facilitating its interaction with a chaperone complex that includes chaperonin-containing TCP1 subunit 5 (CCT5) and promotes the disassembly of primary cilia in human RPE cells [132] (Figure 2C). The C-terminal of CEP55 is critical for both AURKA binding and cilia disassembly [132]. Cep55 KO mice recapitulate many aspects of MARCH, including elongation of primary cilia [132]. These findings suggest that impairment of CEP55–AURKA signaling may play a critical role in the congenital anomalies observed in MARCH.

In contrast, evidence suggest that hyperactivation of CEP55–AURKA signaling may be associated with tumorigenesis. CEP55 expression is increased in human glioma tissues and cell lines compared with normal brain tissue and cells [133,134], and high CEP55 expression in glioma is related to poor prognosis [134]. Consistent with this, suppression of CEP55 in human glioma cell lines decreases proliferation [133,134]. Of note, primary cilia are often downregulated in glioblastoma [109]. Given that AURKA has been proposed as a potential therapeutic target in glioblastoma [25], these findings suggest that inhibition of CEP55–AURKA signaling could be a novel strategy for the treatment of glioma and glioblastoma.

## 3. AKT Signaling and Its Regulation in Primary Cilia

The serine/threonine kinase AKT plays a crucial role in signaling pathways involved in multiple cell functions, including survival, growth, metabolism, proliferation, and differentiation [21]. AKT activation is initiated by engagement of G protein-coupled receptors or receptor tyrosine kinases that are linked intracellularly to class I phosphatidylinositol-3-kinase (PI3K) [21,135]. Activated class I PI3K phosphorylates phosphatidylinositol 4,5-bisphosphate (PI(4,5)P2) at the plasma membrane to generate phosphatidylinositol 3,4,5-trisphosphate (PI(3,4,5)P3), which then binds to the PH domain of AKT. This interaction recruits AKT to the plasma membrane where it is phosphorylated on threonine (Thr)308 and Ser473 by phosphatidylinositol-dependent protein 1 and mammalian target of rapamycin complex (mTORC) 2, respectively [136]. Dual phosphorylation of AKT at these sites fully activates its enzymatic activity and results in phosphorylation of various key substrates, including B-cell lymphoma-2-associated agonist of cell death, Forkhead box O3, tuberous sclerosis complex 1/2, and glycogen synthase kinase 3β (GSK3β) [135]. These AKT substrates are pivotal regulators of many cellular functions, including protein synthesis, autophagy, proliferation, and differentiation [135,136]. With respect to cilia homeostasis, AKT-mediated phosphorylation of GSK3β located at the cilia axoneme suppresses cilia assembly and stability, which contributes to various ciliopathy phenotypes [92,137,138]. The noncatalytic function of AKT is also involved in these activities [6], and AKT activation can occur in a subcellular compartment-specific manner [21,92,135,139,140]. Here, we highlight the regulation of AKT signaling in primary cilia by three mechanisms: by TCHP through altered lipid raft dynamics around primary cilia (Figure 3A), by inositol polyphosphate-5-phosphatase E (INPP5E) (Figure 3B), and by the tumor suppressor von Hippel–Lindau (VHL) protein.

### 3.1. TCHP

TCHP deletion has several effects on lipid metabolism associated with primary cilia in both cultured cells and mice (Figure 3A). While the primary cilia of preadipocytes are normally elongated initially and then gradually shorten during differentiation [141], TCHP KO results in longer than normal primary cilia during differentiation [79]. Various receptors located in and/or around primary cilia are involved in adipogenesis [142], including the insulin receptor (IR) [79], insulin-like growth factor 1 receptor (IGF1R) [143,144,145], Patched-1 and Smoothened [146], and free fatty acid receptor 4 [78]. During adipogenesis, IR/IGF1R–AKT signaling is positively modulated by lipid rafts [147,148], which are membrane nanodomains that regulate multiple cellular functions, including proliferation, differentiation, and apoptosis [149,150,151,152,153,154,155,156]. Lipid rafts act as hubs for recruitment of many signaling proteins, including components of the PI3K–AKT cascade, in response to internal and external stimuli [153,157,158,159,160,161,162]. Studies in the mouse mesenchymal progenitor cell line C3H10T1/2 have shown that exposure of the cells to adipogenic stimuli leads to accumulation of lipid rafts containing caveolin 1 (CAV1) or ganglioside GM3 around the base of primary cilia [79]. TCHP KD in these cells does not affect the localization of IRs but suppresses accumulation of CAV1- or GM3-positive lipid rafts around the ciliary base, which inhibits Akt signaling and disrupts cell differentiation to adipocytes [79]. Notably, TCHP KO mice are resistant to the deleterious metabolic consequences of a high-fat diet [79]. Taken together, these findings suggest that TCHP–AKT signaling may be a novel therapeutic target for the development of anti-obesity agents.

In addition to the accumulation of lipid rafts, TCHP KD in C3H10T1/2 cells reduces the abundance of actin filaments in primary cilia compared with control cells [163]; however, simultaneous KD of intraflagellar transport protein 88, which is required for ciliogenesis [164], ameliorates the effects of TCHP KD on cilia length, actin filamentation, CAV1- or GM3-positive lipid raft accumulation, AKT phosphorylation, and adipogenesis [79,163]. Actin filaments are thought to play important roles in the dynamics of CAV1 [165,166,167]. These findings suggest that TCHP may regulate AKT activity through effects on lipid raft dynamics around primary cilia [79,163].

Modulation of lipid rafts has attracted attention as a promising approach to various diseases in addition to obesity, including cancer and inflammation [162,168,169,170]. In addition to AKT, signaling proteins associated with other cascades, including the mitogen-activated protein kinase pathway and the Janus kinase-signal transducer and activator of transcription pathway, are assembled in lipid rafts and positively and/or negatively regulate signal transduction into the cell [129,133,134,135,136,137,138]. Prostate cancer and melanoma are both associated with a reduction in the abundance of primary cilia [65,67,72,171,172], and hyperactivation of lipid raft–AKT signaling is also observed in these cancers [21,154,156,170,173]. Stimulation of ciliogenesis via inhibition of lipid raft accumulation and suppression of lipid raft–AKT signaling around primary cilia may thus be a potential method for inhibiting the growth of melanoma and prostate cancer [163]. Nevertheless, the relationship between primary cilia and lipid rafts remains to be fully elucidated [174].

### 3.2. INPP5E

The activity of AKT is decreased by dephosphorylation of PI(3,4,5)P3 to PI(4,5)P2 by phosphatase and tensin homolog deleted from chromosome 10 (PTEN) and of PI(3,4,5)P3 to PI(3,4)P2 by several 5′-phosphatases, including inositol polyphosphate-5-phosphatase (INPP5) D, INPP5E, INPP5J, and INPP5K [135,175]. Among these phosphatases, INPP5E is located in primary cilia and is a regulator of AKT signaling at this location [92].

Mutations in INPP5E are associated with Joubert syndrome, a recessive neurodevelopmental ciliopathy that results in underdeveloped and malformed brain structures. The pathogenic mutations in INPP5E decrease the dephosphorylation of PI(3,4,5)P3, resulting in hyperactivation of AKT and suppression of ciliogenesis [176] (Figure 3B). In mice, conditional inactivation of INPP5E in kidney epithelial cells causes hyperactivation of AKT and mTORC1, reduced numbers of primary cilia, and polycystic kidneys [177]. Additionally, deletion of INPP5E in mouse neurons causes aberrant activation of AKT signaling and impairs axon tract development [140]. KD of inpp5e in zebrafish also increases PI(3,4,5)P3 accumulation and suppresses the formation of primary cilia [178].

Mutations in INPP5E are also found in various cancers, including stomach adenocarcinoma, glioblastoma multiforme, and lung adenocarcinoma [179]. Such mutations frequently involve the phosphatase domain [179], suggesting that dysregulation of AKT signaling and primary cilia may contribute to tumor growth. In addition to the regulation of ciliogenesis, INPP5E also controls chromosomal integrity [179]. The mechanisms underlying oncogenesis associated with INPP5E mutations remain largely unknown.

### 3.3. VHL

VHL is an E3 ubiquitin ligase that plays an important role in the cellular response to hypoxia via its regulation of substrates such as the transcription factors hypoxia-inducible factor (HIF) 1α and 2α [180]. Under normoxic conditions, VHL binds to HIF1α that has been hydroxylated at proline (Pro) 402 and/or 564 by the enzymes prolyl-4 hydroxylase domain (PHD) 1, 2, and 3 and subsequently ubiquitinates HIF1α, leading to its degradation [181]. Under hypoxic conditions, however, the activities of PHD1–3 are inhibited, which prevents VHL-mediated ubiquitination and degradation of HIF1α [182] and increases the transcription of hypoxia-related genes.

Mutation of VHL is associated with von Hippel–Lindau syndrome, a rare inherited disorder that causes malignant and benign neoplasms and multiple cysts, especially in the kidney [183,184]. Impairment of VHL in human renal clear cell carcinoma (RCC) has been shown to inhibit the formation of primary cilia [58,137,185]. In mouse embryonic fibroblasts, VHL binds to AKT1 hydroxylated at Pro125 and Pro314 by Phd2, which results in suppression of AKT kinase activity but does not increase its degradation [186,187]. In human RPE cells, loss of primary cilia caused by VHL depletion can be rescued by AKT inhibition [188].

Biallelic inactivation of VHL is the most frequent cause of RCC [189,190,191]. This disease is associated with a severe reduction in the frequency of primary cilia [59] and hyperactivation of the PI3K–AKT signaling cascade [192]. Interestingly, inhibition of AKT in VHL-deficient cells decreases the expression of AURKA [188]. Therefore, inhibition of VHL–AKT signaling may be one approach to suppress the proliferation of RCC through stimulation of ciliogenesis.

## 4. Future Directions

The information reviewed here illustrates how elucidation of the molecular mechanisms underlying signaling by AURKA and AKT associated with primary cilia may provide valuable insights into both the physiological and pathological functions of primary cilia. In turn, these insights lay the foundation for the development of novel therapeutics for cilia-related disorders. Some approaches to drug development may include (i) small molecules that modulate the interaction between kinases and the binding partners that regulate primary cilia, (ii) small molecules that selectively promote degradation of the kinases or their binding partners, and (iii) identifying novel and druggable AURKA and AKT binding partners crucial to their functions in the context of primary cilia.

Intense work over the past few decades has resulted in the development of inhibitors that target ATP-binding sites and/or allosteric sites in multiple kinases, including AURKA and AKT [6,26,193]. More recent advances have enabled the development of agents that interfere with binding of kinases to scaffold proteins that support kinase activation [11,13,24,194,195,196]. During prophase and metaphase, AURKA is recruited to microtubules in mitotic spindles through the interaction between the C-terminal catalytic domain of AURKA and the N-terminus of TPX2 [197,198]. As noted earlier, this interaction is crucial for regulating the phosphorylation state and activity of AURKA, and, importantly, it is also druggable [194]. The small molecule AURKA inhibitor AurkinA acts by binding to the hydrophobic pocket of AURKA where TPX2 is normally accommodated through a conserved Tyr-Ser-Tyr motif in TPX2 [11]. Binding of AurkinA causes mislocalization of AURKA from the microtubules in mitotic spindles and inhibits its catalytic activity without affecting ATP binding [11]. Novel approaches are currently being developed to find chemical spaces in AURKA that can modulate its protein–protein interactions [199,200,201].

Technologies that lead to targeted protein degradation, such as proteolysis-targeting chimeras (PROTACs), protein-catalyzed capture agents (PCCs), and specific and nongenetic inhibitors of apoptosis protein-dependent protein erasers, have been successfully applied to develop novel kinase inhibitors [202,203,204,205,206]. For example, a PROTAC consisting of alisertib, a clinically used ATP-competitive inhibitor of AURKA [207], and thalidomide, which induces protein degradation via cereblon-containing ubiquitin ligase, results in proteolysis of AURKA and arrest of the cell cycle in MV4-11 human acute myeloid leukemia cells [5]. AKT can also be selectively degraded by a PROTAC that utilizes ipatasertib and lenalidomide as the ATP-competitive AKT inhibitor and degradation inducer, respectively [208]. Similarly, AKT degradation is induced by a PCC that employs a peptide derived from human immunodeficiency virus type 1 Tat protein as the cell-penetrating peptide and an HIF1α peptide as a VHL ligand [209]. In addition to the kinases themselves, protein degradation systems have also been designed to target kinase modulators. The protein bromodomain containing 4 (BRD4) binds to P-TEFb, a heterodimer of cyclin-dependent kinase 9 (CDK9) and cyclin T1 and promotes transcriptional elongation through phosphorylation of RNA polymerase II [210,211]. A PROTAC composed of pomalidomide linked to OTX-015, a small molecule that binds to BRD4 at the bromodomain and extra-terminal domain, induces BRD4 degradation and consequently decreases the activity of CDK9 and expression of its downstream target, MYC [212]. Thus, targeted protein degradation can be applied to develop novel inhibitors of protein–protein interactions [213].

Subcellular compartment-specific signaling can be analyzed using Förster or fluorescence resonance energy transfer (FRET) sensors fused to signal peptides. FRET sensors have been developed to analyze multiple signaling pathways, including those involving AURKA [214], AKT [139,215], cyclic adenosine monophosphate (AMP) [216,217], and calcium [218]. For example, an AURKA FRET sensor composed of AURKA within an eGFP and mCherry donor–acceptor fluorophore pair was based on the conformational change exhibited by AURKA upon autophosphorylation of Thr288 [214]. Phosphopeptide-binding domains (PBDs) have also been employed in the development of FRET biosensors to visualize kinase activity [219]. For example, Eevee-iAkt, a FRET biosensor for AKT, is composed of YPet as the acceptor fluorophore, the Forkhead-associated domain of yeast Rad53 as the PBD, an optimized peptide derived from human GSK3β as the AKT substrate sequence, and eCFP as the donor fluorophore [215]. Addition of the C-terminal region of human H-Ras and K-Ras to the C-terminal of Eevee-iAkt localizes the expression of Eevee-iAkt to raft and nonraft domains, respectively, and enables AKT activity in each domain to be analyzed [215]. Calcium and cyclic AMP signaling in primary cilia have also been successfully analyzed using FRET sensors. One example that has facilitated analysis of calcium signaling in primary cilia is composed of calmodulin as the calcium-binding domain, M13 as the calcium-bound calmodulin-binding domain, and eCFP and YPet as the donor and acceptor fluorophores, respectively. This sensor can be selectively expressed in primary cilia by linkage to the ciliary protein ADP ribosylation factor-like GTPase 13B (ARL13B) [218]. Similarly, fusion of ciliary proteins such as ARL13B with the adenylyl cyclase-coupled somatostatin receptor 3 has been employed to construct FRET sensors for analysis of cyclic AMP signaling in primary cilia [217,220]. Based on these studies, it seems likely that FRET sensors could also be constructed for the analysis of AURKA and AKT signaling localized in primary cilia. A complete picture of the interactomes of subcellular compartments, including primary cilia and lipid rafts, is gradually being deciphered by studies using other novel approaches, such as proximity mapping and stable isotope labeling using amino acids in cell culture [221,222,223]. These approaches have been successfully used to reveal the interactome of AURKA [27] and AKT [224].

Thus, a combination of these techniques is likely to advance our understanding of the mechanisms regulating AURKA and AKT kinase signaling within and around primary cilia and may pave the way for the development of novel therapeutics for ciliopathies.

## Figures and Tables

**Figure 1 cells-10-03602-f001:**
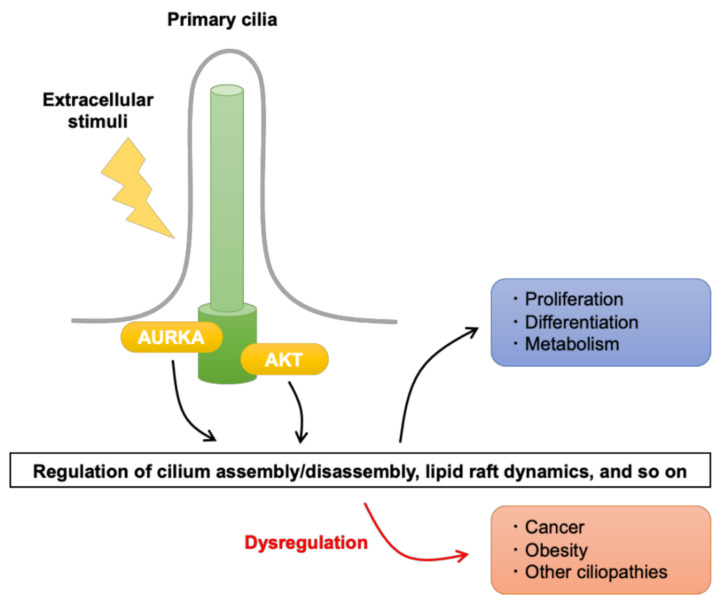
Overview of the involvement of AURKA and AKT associated with primary cilia in cellular functions. AURKA and AKT located at the ciliary base mediate signaling from extracellular stimuli that regulate crucial cellular functions, including proliferation, differentiation, and metabolism. Among other functions, AURKA and AKT signaling regulates the assembly and disassembly of primary cilia and the dynamics of signaling hubs known as lipid rafts, which are located in the plasma membrane around primary cilia. Dysregulation of these functions contributes to a number of ciliopathies, including cancer and obesity, as described in this review.

**Figure 2 cells-10-03602-f002:**
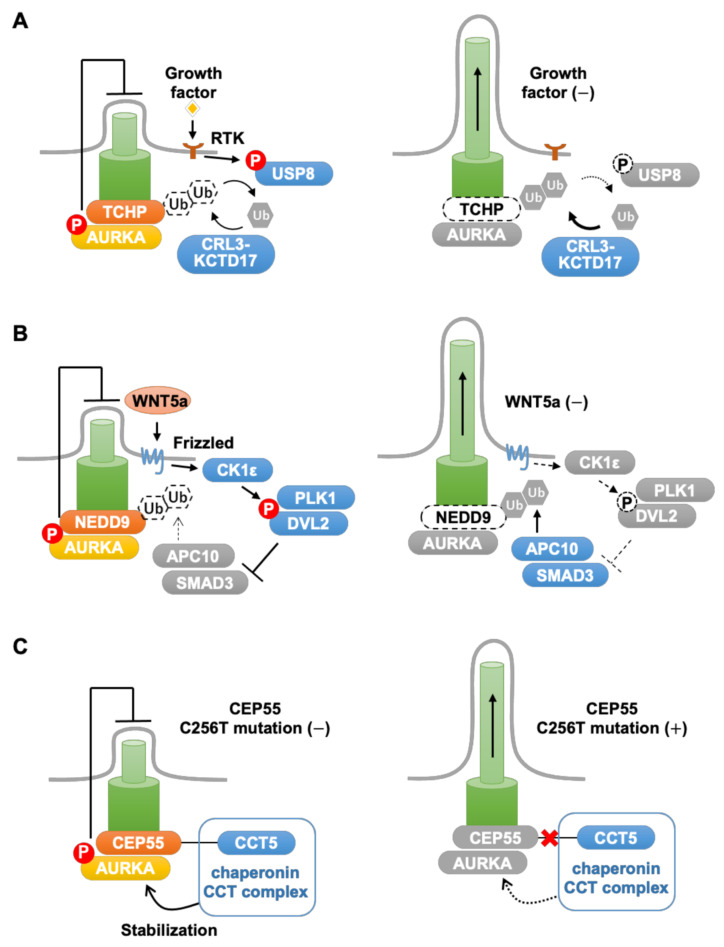
AURKA signaling associated with primary cilia. (**A**) Regulation of AURKA through TCHP. In the presence of growth factors (left panel), USP8 is activated by RTK-mediated phosphorylation, leading to deubiquitination of TCHP, activation of AURKA, and suppression of ciliogenesis. In the absence of growth factor (right panel), USP8 is inactive, and TCHP is degraded via ubiquitination by CRL3-KCTD17. (**B**) Regulation of AURKA through NEDD9. In the presence of WNT5a (left panel), CK1ε phosphorylates DVL2, resulting in suppression of NEDD9 ubiquitination through APC10. NEDD9 activates AURKA and suppresses ciliogenesis. In the absence of WNT5a (right panel), NEDD9 is targeted for degradation by APC10-mediated ubiquitination. (**C**) CEP55 stabilizes AURKA. The CCT5-containing chaperonin CCT complex interacts with wild-type CEP55 and stabilizes AURKA, resulting in suppression of ciliogenesis (left panel). In cells harboring a Cys256Thr mutation in CEP55, which is associated with MARCH, mutant CEP55 fails to localize to the centrosome, leading to destabilization of AURKA and elongation of primary cilia (right panel). Abbreviations: APC10, anaphase-promoting complex subunit 10; CCT5, chaperonin-containing TCP1 subunit 5; CK1ε, casein kinase 1ε; CRL3-KCTD17, E3 ligase complex composed of cullin 3, ring-box 1, and potassium channel tetramerization domain–containing 17; Cys, cysteine; DVL2, disheveled segment polarity protein 2; MARCH, multinucleated neurons, anhydramnios, renal dysplasia, cerebellar hypoplasia, and hydranencephaly; PLK1, polo-like kinase 1; RTK, receptor tyrosine kinase; SMAD3, SMAD family member 3; Thr, threonine; Ub, ubiquitin; USP8, ubiquitin-specific protease 8; WNT5a, Wnt family member 5A.

**Figure 3 cells-10-03602-f003:**
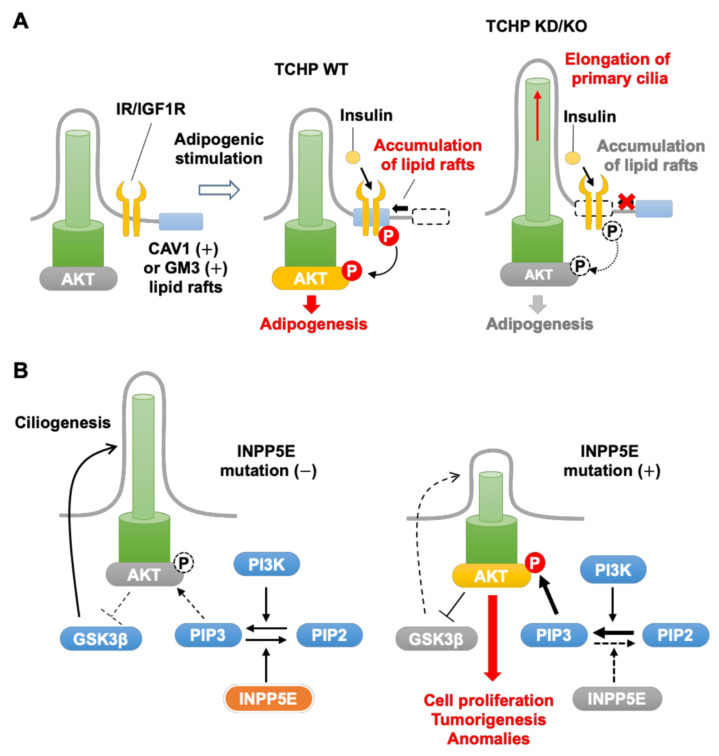
AKT kinase signaling associated with primary cilia. (**A**) Lipid raft-mediated activation of AKT at the base of primary cilia during adipogenesis. Exposure of preadipocytes to adipogenic stimuli activates IR and IGF1R located at the ciliary base and leads to accumulation of CAV1- or GM3-positive lipid rafts, phosphorylation of AKT, and promotion of adipogenesis. KD or KO of Tchp elongates primary cilia of preadipocytes, which inhibits the accumulation of lipid rafts upon adipogenic stimulation. (**B**) Regulation of AKT by INPP5E. PI3K and INPP5E balance the generation of PIP3, a key activator of AKT. Loss-of-function mutation in INPP5E increase PIP3, hyperactivates AKT, and suppresses ciliogenesis, leading to cell proliferation, tumorigenesis, and anomalies. Abbreviations: CAV1, caveolin 1; GSK3β, glycogen synthase kinase 3β; INPP5E, inositol polyphosphate-5-phosphatase E; IGF1R, insulin-like growth factor 1 receptor; IR, insulin receptor; PI3K, phosphatidylinositol-3-kinase; PIP2, phosphatidylinositol 4,5-bisphosphate; PIP3, phosphatidylinositol 3,4,5-triphosphate.

## Data Availability

Not applicable.

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
