# Peer review of "Aurora A and AKT Kinase Signaling Associated with Primary Cilia"

_cells, 2021, doi:10.3390/cells10123602_

Round 1
Reviewer 1 Report
The manuscript entitled “Aurora A and AKT kinase signaling associated with primary cilia” by Nishimura et al. summarizes the recent literature about the role of the Ser/Thr kinases Aurora A and AKT in primary cilia signaling. Dysregulation of kinase signaling is associated with various pathological conditions. However, the underlying molecular mechanisms are often ill-defined. Kinase signaling is often organized in subcellular compartments. Here, primary cilia are a prime example, as they function as a signaling hub that transduces information from the environment into a cellular response. Primary cilia dysfunction leads to severe diseases, commonly referred to as ciliopathies. Aurora A and AKT have been shown to control primary cilia signaling. Here, the authors summarize the recent knowledge about the role of these kinases in the development of ciliopathies.
However, the manuscript contains a number of issues that need to be fixed before being considered for publication. Thus, the following points need to be addressed:
- The review would greatly benefit from schematic overviews that illustrate the ciliary signaling pathways, which are described in the different chapters. Please add these overviews.
- In the introduction, the first paragraph is not well connected to the 2nd and 3rd The first paragraph ends with the statement that the kinases play an important role in primary cilia. But the next paragraphs deal with the role of the kinases in melanoma and the general cellular functions of these enzymes. Thus, the connection between cancer and primary cilia signaling and function is not well outlined in the introduction. Please rewrite the induction and provide a schematic overview of the model of how primary cilia dysfunction might cause cancer development.
- In general, the cilia literature that is cited by the authors by large consists of reviews and not original publications. Please add original publications for the different topics.
- The connection between obesity and primary cilia dysfunction is not introduced at all – please add.
- The connection between cilia dynamics and cell cycle regulation is not well explained. Please add a schematic overview (as a joined figure with 2)).
- In general, many results that are described are not well introduced and the connection is not clear (line 165: PKD not introduced, line 169: description of BBS proteins – what do they do? etc.). In the current state, the manuscript is not suited for a more general cell biology audience but rather for a specialized, primary cilia audience.
- Some abbreviations have not been explained.
- It is not clear to the reader where the signaling takes place. Are all aspects described here happening in the cilium or at the ciliary base or are some of them even occurring in the cell soma? Please clarify the spatial organization of the signaling pathways in the different chapters.
- Tables 1 and 2 need to be graphically improved – the bullet points are not aligned and the appearance, in general, is not great.
- The outlook is also quite confusing with respect to how the different aspects are explained as not enough information is provided.
Reviewer 2 Report
The review by Nishimura et al “Aurora A and AKT kinase signaling associated with primary 2 cilia” is suitable for the publication in Cells.
There are only some minor comments:
- Line 88: Please add that Aurora A is also associated with centrosome separation.
- I miss a part about Calmodulin, Aurora A and NEDD9.
- And about obesity, the primary cilium and Aurora A. There are also works describing that the function of the primary cilium of obese adipose-derived mesenchymal stem cells is rescuable by low-dose inhibition of Aurora A or Erk1/2 (for example Ritter et. al, Stem Cell Res Ther, 2019).
Reviewer 3 Report
The review is well structured and written, with comprehensive list of references about the topic. You might consider reformulating the abstrast and title to better fit the review contents, as it ( at least from my point of view) does not deal so much with AURKA and AKT involvement in ciliopathies as with the AURKA and AKT in primary ciliogenesis and dependent signaling as such.
The tables 1 and 2 need reformatting to fit better, now they are not clear at the first or even a second glance.
Round 2
Reviewer 1 Report
All issues have been addressed.